# Novel 1,3,4-Oxadiazole Derivatives Containing a Cinnamic Acid Moiety as Potential Bactericide for Rice Bacterial Diseases

**DOI:** 10.3390/ijms20051020

**Published:** 2019-02-26

**Authors:** Shaobo Wang, Xiuhai Gan, Yanju Wang, Shaoyuan Li, Chongfen Yi, Jixiang Chen, Fangcheng He, Yuyuan Yang, Deyu Hu, Baoan Song

**Affiliations:** State Key Laboratory Breeding Base of Green Pesticide and Agricultural Bioengineering/Key Laboratory of Green Pesticide and Agricultural Bioengineering, Ministry of Education, Guizhou University, Huaxi District, Guiyang 550025, China; wangshaobo97@126.com (S.W.); 18985900557@163.com (Y.W.); 15761626670@163.com (S.L.); 13608581558@163.com (C.Y.); 18230826028@163.com (J.C.); 13667177168@163.com (F.H.); gs.yuyuanyang17@gzu.edu.cn (Y.Y.); dyhu@gzu.edu.cn (D.H.)

**Keywords:** 1,3,4-oxadiazole derivatives, cinnamic acid, bacterial diseases, antibacterial activity, 3D-QSAR, real-time quantitative PCR

## Abstract

Rice bacterial leaf blight and leaf streak are two important bacterial diseases of rice, which can result in yield loss. Currently, effective antimicrobials for rice bacterial diseases are still lacking. Thus, to develop highly effective and low-risk bactericides, 31 novel 1,3,4-oxadiazole derivatives containing a cinnamic acid moiety were designed and synthesized. Bioassay results demonstrated that all compounds exhibited good antibacterial activities in vitro. Significantly, compounds **5r** and **5t** showed excellent antibacterial activities against *Xanthomonas oryzae* pv. *oryzae* (*Xoo*) and *X. oryzae* pv. *oryzicola* (*Xoc*), with the 50% effective concentration (EC_50_) values of 0.58 and 0.34, and 0.44 and 0.20 μg/mL, respectively. These compounds were much better than thiodiazole copper (123.10 and 161.52 μg/mL) and bismerthiazol (85.66 and 110.96 μg/mL). Moreover, compound **5t** had better protective and curative activities against rice bacterial leaf blight and leaf streak than thiodiazole copper and bismerthiazol in vivo. Simultaneously, the in vivo efficacy of the compounds was demonstrated by real-time quantitative PCR to quantify bacterial titers. In addition, a three-dimensional quantitative structure–activity relationship model was created and presented good predictive ability. This work provides support for 1,3,4-oxadiazole derivatives containing a cinnamic acid moiety as a potential new bactericide for rice bacterial diseases.

## 1. Introduction

As the main food crop for more than half of the world’s population, rice is susceptible to a large number of pathogenic microorganisms, which can cause devastating diseases and result in serious loss of production worldwide, threatening global food security [1,2,3]. In rice-growing countries, rice bacterial leaf blight and leaf streak are two important devastating bacterial diseases of rice; they are caused by the pathogens *Xanthomonas oryzae* pv. *oryzae* (*Xoo*) and *X. oryzae* pv. *oryzicola* (*Xoc*), respectively [4,5]. These two kinds of bacterial diseases can occur either individually or collectively, and over the past 30 years, each of them has been shown to result in yield loss of at least 10% of susceptible rice varieties grown on a large scale [4]. The hazards are increasing with the annual expansion of planting areas. Currently, few bactericides, such as bismerthiazol, thiodiazole copper, zinc thiazole, and zhongshengmycin are available for controlling rice bacterial diseases. However, rice bacterial leaf blight and leaf streak have not yet been effectively controlled, and the intensive and continuous use of the same bactericide has led to the development of drug-resistant pathogens in the bacterial population [6,7,8]. Therefore, developing new, highly effective bactericides for controlling the two bacterial diseases of rice is an imperative and daunting task.

Over the past few decades, natural products have been used in the development of new pesticides, given their advantages of easy decomposition and environmental friendliness. As one of the natural aromatic fatty acids, cinnamic acid is the deaminated product of phenylalanine in plant tissues [9]. Substituted cinnamic acids show diverse biological activities, such as antitumor [9], antioxidation [10,11], insecticidal [12], antiviral [13,14], fungicidal [15,16], bactericidal [17], and herbicidal activities [18], and have attracted the attention of many researchers. In particular, many cinnamic acid analogs are crucial and promising compounds as potential antifungal and antibacterial agents [19]. Reportedly, cinnamic acid has been shown to have an excellent bactericidal effect in combination with fumaric acid [20].

1,3,4-Oxadiazole derivatives are an important class of nitrogen-containing heterocyclic compounds and are used as pesticides because of their antibacterial [21,22,23], antifungal [24], nematicidal [25], antiviral [26], herbicidal [27], and insecticidal activities [28]. In our previous works, we found that some novel 1,3,4-oxadiazole derivatives containing the sulfone moiety showed potent antibacterial and antifungal activities [21,22,23,24]. Among them, compounds **a** and **b** displayed excellent activities for controlling rice bacterial leaf blight and leaf streak diseases, as shown in Figure 1. The relationship between the structure and activity confirmed that the 1,3,4-oxadiazole–sulfone backbone is a vital pharmacophore.

To continue the research on 1,3,4-oxadiazole derivatives, we set out to create novel 1,3,4-oxadiazole derivatives containing a cinnamic acid moiety replacing the methylene group with a vinyl group in the present work, as shown in Figure 1. In addition, we tested their antibacterial activities for rice bacterial leaf blight and leaf streak in vitro and in vivo. Moreover, the in vivo efficacy of the compounds was confirmed by real-time quantitative PCR assays. We also established a three-dimensional quantitative structure–activity relationship (3D-QSAR) model.

## 2. Results

### 2.1. Synthesis

The pathways for the synthesis of the 1,3,4-oxadiazole derivatives containing a cinnamic acid moiety are described and shown in Scheme 1. The different substituted cinnamic acids were used as the starting materials via hydrazidation, cyclization, thioetherification, and oxidation reaction to obtain the target compounds.

### 2.2. In Vitro Antibacterial Activity

#### 2.2.1. In Vitro Antibacterial Activity of *Xoo*

The title compounds **5a**–**5ae** were tested for in vitro antibacterial activities against rice bacterial leaf blight caused by *Xoo* via the turbidimeter test. The results are shown in Table 1.

As shown in Table 1, all the title compounds showed better bactericidal activities against *Xoo* than the commercial agents bismerthiazol and thiodiazole copper. Among them, compounds **5a**, **5b**, **5d**, **5k**, **5n**, **5q**, **5r**, **5s**, **5t**, **5w**, **5y**, and **5ae** showed good antibacterial activities with inhibition rates of 100% at 50 μg/mL concentration. The compounds **5q**, **5r**, and **5t** showed excellent antibacterial activities with inhibition rates of 100%, 97.3%, and 99.8%, respectively, at 5 μg/mL concentration. In addition, the 50% effective concentration (EC_50_) values of the target compounds **5a**–**5ae** were tested. As shown in Table 1, all the target compounds showed better antibacterial activity than thiodiazole copper (123.10 μg/mL), bismerthiazol (85.66 μg/mL), and cinnamic acid (236.07 μg/mL). Compounds **5r** and **5t** showed the most obvious antibacterial activity with EC_50_ values of 0.58 and 0.44 μg/mL, respectively, which were higher than those of leading compounds **a** (1.41 μg/mL) and **b** (0.84 μg/mL).

#### 2.2.2. In Vitro Antibacterial Activity of *Xoc*

The in vitro antibacterial activities of the title compounds **5a**–**5ae** against *Xoc* were evaluated using the turbidimeter test. As shown in Table 2, all the target compounds revealed better bactericidal activities against *Xoc* than bismerthiazol and thiodiazole copper. Among them, compounds **5k**, **5m**, **5n**, **5q**, **5r**, **5s**, and **5t** had good antibacterial activity with inhibition rates of 100% at 50 μg/mL concentration. Compounds **5k**, **5r**, and **5t** showed excellent antibacterial activity with inhibition rates of 92.5%, 96.6%, and 98.2% at 5 μg/mL, respectively. Furthermore, the EC_50_ values of the target compounds **5a**–**5ae** were tested, and all of them showed better antibacterial activity against *Xoc* than thiodiazole copper (161.52 μg/mL) and bismerthiazol (110.96 μg/mL). Compounds **5r** and **5t** showed excellent antibacterial activity with EC_50_ values of 0.34 and 0.20 μg/mL, respectively, which were higher than those of compounds **a** (2.01 μg/mL) and **b** (1.16 μg/mL).

### 2.3. In Vivo Antibacterial Activity

#### 2.3.1. In Vivo Antibacterial Activities Against *Xoo*

As shown in Figure 2a and Table 3, at the concentration of 200 μg/mL, the target compound **5t** had good in vivo protective activities against rice bacterial leaf blight. The control efficiency was 66.62%, which was superior to those of thiodiazole copper (57.33%) and bismerthiazol (51.71%). Figure 2b and Table 4 reveal that compound **5t** showed excellent in vivo curative activity against rice bacterial leaf blight with control efficiencies of 56.05%, respectively, which was better than those of bismethylthiazole (43.65%) and thiazole copper (48.60%).

#### 2.3.2. In Vivo Antibacterial Activities Against *Xoc*

As shown in Figure 3a and Table 5, the target compound 5t exerted good in vivo protective activity (58.22%) against rice bacterial leaf streak at 200 μg/mL concentration, which was better than thiodiazole copper (53.77%) and bismerthiazol (46.08%). Meanwhile, Figure 3b and Table 6 show that compound 5t had good in vivo curative activity against rice bacterial leaf streak. The control efficiency was 55.92%, which was higher than those of thiodiazole copper (42.83%) and bismerthiazol (45.25%).

### 2.4. Real-Time Quantitative PCR Assays

The fluorescence amplification curve (Figure 4a,d), melting curve (Figure 4b,e) and standard curve (Figure 4c,f) of known concentration *Xoo* and *Xoc* cultures are shown in Figure 4. Melting curve analysis showed a single peak for each primer at around 88 °C and 85 °C (Figure 4b,e) suggesting the absence of primer dimers. The standard curve of *Xoo* and *Xoc* (Figure 4c,f) is y = −2.001x + 41.446 and y = −1.8829x + 40.694, and the *R*^2^ is 0.9888 and 0.9822, respectively.

The *Xoo* and *Xoc* titer in rice leaves are shown in Table 7 and Table 8 according to the treatment sample threshold cycle value and the standard curve. As shown in Table 7, the titer of *Xoo* in the rice leaves was 2.42 × 10^11^ CFU/mL after with the compound **5t** in protective treatment, which was lower than negative control (1.04 × 10^12^ CFU/mL) and positive controls (thiodiazole copper: 3.05 × 10^11^ CFU/mL, bismerthiazol: 3.42 × 10^11^ CFU/mL), and the titer of *Xoo* in the rice leaves was 3.05 × 10^11^ CFU/mL in curative treatment, which was lower than that of the negative (1.26 × 10^12^ CFU/mL) and positive (thiodiazole copper: 3.69 × 10^11^ CFU/mL, bismerthiazol: 4.83 × 10^11^ CFU/mL) controls.

As shown in Table 8, in the protective and curative experiment, the titer of *Xoc* in the rice leaves treated with compound **5t** were 2.55 × 10^10^ CFU/mL and 7.35 × 10^10^ CFU/mL, respectively, which were lower than the negative controls (1.20 × 10^11^ and 1.41 × 10^11^ CFU/mL) and the positive controls (thiodiazole copper: 3.83 × 10^10^ and 8.31 × 10^10^ CFU/mL, bismerthiazol: 4.16 × 10^10^ and 7.98 × 10^10^ CFU/mL).

### 2.5. Study of 3D-QSAR Models

The bioactivity data for title compounds against *Xoo* are shown as pEC_50_ (Table 9) and used for 3D-QSAR analysis. The training set of the comparative molecular field analysis (CoMFA) and comparative molecular similarity indices analysis (CoMSIA) models were generated. This training set contained 25 compounds by selecting randomly from all title compounds, whereas the remaining six (asterisk-labeled) compounds were used as the testing set. The SYBYL-X 2.1 software (Tripos Inc., St. Louis, MO, USA) was used to carry out molecular modeling. The Gasteiger–Hückel charge, Tripos force field, and Powell conjugate gradient algorithm were used to optimize the structure. The convergence criterion was 0.005 kcal/mol. For the CoMFA and CoMSIA modeling studies, the 3D structure of 31 molecules was aligned on a common template molecule with **5t**, which has the best bactericidal activity against *Xoo*. The alignment result is shown in Figure 5.

The pEC_50_ value and the parameters of the models were analyzed by partial least squares (PLS) regression. First, leave-one-out cross-validation was performed on the training set compounds. The optimal number of components (ONCs) and the cross-validation coefficient (*q*^2^) were obtained in this step. Then, no validation was performed to verify the fitting ability and other properties of the model. The non-cross-validated correlation coefficient (*r*^2^), *F* value, standard error of estimate (SEE) value, and the field-to-activity contribution value were acquired in the terminal panel when the PLS regression was completed.

Structure–activity relationship was discussed by utilizing the 3D-QSAR models created by CoMFA and CoMSIA. The CoMFA and CoMSIA models were constructed using the bioactivity data of *Xoo*. The relevant parameters of the model are listed in Table 10. Generally, the internal validation of cross-validated *q*^2^ and the non-cross-validation coefficient *r*^2^ values represent the robustness and predictive ability of the 3D-QSAR model. In addition, a satisfactory model has a commonly recognized value of *q*^2^ > 0.5 and *r*^2^ > 0.8. Table 10 shows that the *q*^2^ of the CoMFA model was 0.725, the corresponding *r*^2^ was 0.963, the SEE was 0.114, and the *F* value was 78.624. The *q*^2^ of the CoMSIA model was 0.707, the *r*^2^ was 0.941, the SEE was 0.144, and the *F* value was 48.186. This result shows that the model had a good predictive ability. As shown in Table 10, the contributions of steric and electrostatic fields to the CoMFA model were 65.1% and 34.9%, respectively, indicating that bioactivity was mainly determined by steric interaction. The steric, electrostatic, hydrophobic, H-bond donor, and H-bond acceptor contributions of the CoMSIA models were 15.6%, 32.9%, 47.2%, 0.0%, and 4.2%, respectively, indicating that the target compound biological activity was mainly affected by the hydrophobic and electrostatic effects.

Meanwhile, Table 9 illustrates the actual pEC_50_ of the model and the predicted pEC_50_. The residual between the actual biological activity of the compound and the predicted biological activity is very small, indicating that the model has certain predictive ability. Figure 6 shows the linear relationships between the actual value and the predicted value. The *r*^2^ of the CoMFA and CoMSIA models are 0.9633 and 0.9416, respectively. All data are concentrated near the straight line, indicating that the model has good correlation.

Figure 7 shows the contour maps for the CoMFA model. The steric CoMFA map (Figure 7a) shows a green contour and indicates that the bulky groups would increase the activity. By contrast, yellow indicates the contour regions where the small groups would increase the activity. Therefore, a green block is present in the para position of the benzene ring, which indicates that a large substituted group should be introduced at this position. On the contrary, the yellow contour indicates that the introduction of small groups at this position is advantageous for enhancing the biological activity. Compounds having 4-F, 4-Cl, and 4-Br introduced at position R^1^ are superior to those without substituents, and for the halogen at the R^1^ position, a regularity of 4-Br > 4-Cl > 4-F, for example, **5t** > **5r** > **5n** > **5b**, is present. Moreover, ethyl activity is introduced at position R^2^ better than *n*-pentyl, such as **5b** > **5h**. As shown in the electrostatic CoMFA map (Figure 7b), a red module is present at the para position of the benzene ring, which indicates that the introduction of the negative-charged group is conducive to improve the biological activity, and a blue module is present near other positions of the benzene ring. For example, the introduction of 4-F compound **5n** is more active than the compound **5j**.

The R^1^ position in the steric field of the CoMSIA model (Figure 8a) has roughly the same effect on the biological activity of the compound as that of the CoMFA model. Moreover, the R^2^ position elaborates the relationship of the biological activity of the compound in further detail. The green and yellow modules near the sulfone group indicate that the R^2^ position shows a tendency of increasing first and then decreasing as the group increases in activity, for example, **5a** < **5c**, **5h** < **5c**, and **5w** < **5c**. As shown by the effects of the electrostatic field from the CoMSIA model (Figure 8b), the red module indicates that the introduction of a strong electronegative group will enhance the activity, whereas the blue module indicates that the introduction of a positive group is conducive to activity enhancement, for example, **5y** > **5w**. The hydrophobic CoMSIA map (Figure 8c) shows that the yellow portion shown at the R^1^ position indicates that the introduction of a hydrophilic group favors activity, while the gray portion indicated by the R^2^ position indicates that introduction of a hydrophobic group facilitates activity. The H-bond acceptor CoMSIA map (Figure 8d) shows the purple part and indicates that the addition of a hydrogen bond acceptor favors activity, whereas the red outline indicates that an increase of hydrogen bond donors does not favor activity. The introduction of a hydrogen acceptor at the red moiety near the benzene ring will decrease the biological activity, whereas the introduction of a hydrogen acceptor at the oxadiazole purple moiety increases the biological activity. This also explains why compound **5q** shows better bactericidal activity than **5u**.

## 3. Discussion

In our present work, an unsaturated double bond was added between the benzene ring and the 1,3,4-oxadiazole, that is, combining 1,3,4-oxadiazole with cinnamic acid to form novel 1,3,4-oxadiazole derivatives containing a cinnamic acid moiety. This bond can enhance the antibacterial activities to introduce the double bond in 1,3,4-oxadiazole derivatives. Compounds **5r** and **5t** showed excellent antibacterial activities against *Xoo* and *Xoc* in vitro with EC_50_ values of 0.58 and 0.34, and 0.44 and 0.20 μg/mL, respectively, which were better than leading compounds cinnamic acid (236.07 and 270.45 μg/mL), **a** (1.41 and 2.01 μg/mL), and **b** (0.84 and 1.16 μg/mL). Moreover, compound **5t** had good in vivo protective and curative activity against rice bacterial leaf blight and rice bacterial leaf streak. Meanwhile, the in vivo efficacy of the compounds was demonstrated by real-time quantitative PCR to quantify bacterial titers. In addition, the 3D-QSAR model showed satisfactory predictive ability, and the results illuminated (as shown in Figure 9) that bulky groups, negatively charged groups, hydrophilic groups at R^1^ position, and small groups, positive groups, hydrophobic groups, H-bond acceptor groups at the R^2^ position favor activity. Meanwhile the sulfone group is the essentially active group. These results provide further valuable clues for the design of novel 1,3,4-oxadiazole derivatives containing a cinnamic acid moiety.

## 4. Materials and Methods

### 4.1. Materials and Instrument

Cinnamic acids, 1-Ethyl-3-(3-dimethylaminopropyl) carb-odiimide hydrochloride (EDCI) and 1-hydroxybeonztriazole (HOBt) were purchased from Shanghai Tansoole Chemicals Co., Ltd (Shanghai, China). Other reagents and solvents were purchased from Guiyang Yuda Chemical Reagent Co., Ltd (Guizhou, China). All reagents and solvents were reagent grade and used without further purification. 

Melting points were determined using an X-4 micro digital melting point (Beijing Tech Instruments Co., Beijing, China), and readings were uncorrected. ^1^H NMR and ^13^C NMR spectra were recorded on a JEOL-500 NMR spectrometer (JEOL, Tokyo, Japan) with DMSO-d_6_ or CDCl_3_ as solvent and TMS as internal standard. High-resolution mass spectra (ESI TOF (+)) were measured on a Thermo Fisher Scientific LTQ Orbitrap XL (Waltham, MA, USA). The real-time quantitative PCR experiment was carried out on the instrument of BIO-RAD iCycle which was purchased from Bio-Rad Laboratories, Inc (Hercules, MO, USA). The CoMFA and CoMSIA models were analyzed using SYBYL-X 2.1 (Tripos Inc., St. Louis, MO, USA).

### 4.2. Chemistry

#### 4.2.1. General Procedure for the Synthesis of Intermediates 2 and 3

Intermediate **2** was prepared according to the known method [29]. Intermediate **3** was prepared according to the following method. Carbon disulfide (0.75 mol) was slowly added dropwise to a three-necked flask containing KOH (0.6 mol) of ethanol (500 mL) at room temperature to form potassium ethylxanthate. Then, intermediate **2** (0.5 mol) was added, and the reaction was heated to 40–50 °C and heated to reflux. After the reaction was completed, the reaction mixture was spin-dried to remove the solvent ethanol to give a dark-brown powdery solid in 85–90% yields.

#### 4.2.2. General Procedure for the Synthesis of Intermediate 4

Intermediate **3** (1.0 mmol) and potassium carbonate (3.0 mmol) were added to a three-necked flask containing dimethylformamide (DMF) 8 mL and stirred at room temperature. Then, R_2_X (1.1 mmol) was slowly added dropwise. The mixture was heated to 70–80 °C and stirred continuously for 3–5 h. After the reaction was completed and cooled to 20–25 °C, the reaction solution was dropped into water, stirred, and dispersed, and filtered to obtain a yellow solid powder intermediate **4**.

#### 4.2.3. General Procedure for the Title Compounds 5a–5ae

The title compounds were prepared following the reported methods [30]. Intermediate **4** (1 mmol) was dissolved in ethanol (10 mL). Then, a solution of ammonium molybdate (0.2 mmol) in 30% hydrogen peroxide (20 mmol) was added and stirred for 5 h (as monitored by TLC). The solvent was removed under vacuum, and a saturated aqueous solution of sodium bicarbonate was added to adjust the pH to 8–9. Then, the mixture was extracted with CH_2_Cl_2_, and the solvent was dried and evaporated under reduced pressure to obtain a yellow solid. The crude product was also recrystallized from ethanol to obtain the pure target compounds **5a**–**5ae** in 48.9–89.3% yields. The physical, NMR, and HRMS data of title compounds were listed in Appendix A.

### 4.3. In Vitro Antibacterial Activity

The title compounds **5a**–**5ae** were determined for in vitro antibacterial activities against *Xoo* and *Xoc* using the turbidimeter test [31]. The test solutions of 50 and 5 μg/mL were prepared by dissolving the title compound in dimethylsulfoxide (DMSO) and diluting with 0.1% Tween-20 in sterile distilled water. Thiodiazole copper (20% suspension concentrate) and bismerthiazol (20% wettable powder) were used as a positive control. The DMSO was used as a negative control. Approximately 4 mL of nutrient broth (NB) media was added with 1 mL test solution and 40-μL culture solutions containing the pathogens of *Xoo* or *Xoc*. The inoculated tubes were incubated for 36–48 h at 180 rpm and 28 ± 1 °C in a constant temperature shaker. Cultures were monitored for growth on the microplate reader by measuring the optical density at 595 nm (OD 595) until the bacteria in the untreated nutrient broth (NB) media were in logarithmic growth.
Inhibition rate (%) = (untreated − treated)/untreated × 100 (1)

The results of antibacterial activities of the target compounds against *Xoo* and *Xoc* were expressed by EC_50_ and were evaluated by SPSS17.0 software. The experiment was repeated three times for each compound.

### 4.4. In Vivo Antibacterial Activity

#### 4.4.1. In Vivo Antibacterial Activities Against Rice Bacterial Leaf Blight

Potted plants were used to determine the in vivo protection and curative activity of compounds under greenhouse conditions according to previously reported methods [23]. Target compound **5t** was tested for their protective activities. Bismerthiazol and thiodiazole copper were used as positive controls. After the “Fengyouxiangzhan” rice seeds were sowed at approximately one and a half months, the test compound dissolved in DMSO was diluted to a concentration of 200 μg/mL by 0.1% Tween-20 water. Sterile distilled water was used in the negative control. The liquid water was evenly sprayed on rice leaves. After one day, a 2–3 cm section of the rice leaf tip was cut, and the wound site was soaked in the bacterial solution for 5 s. At the same time, 10 leaves were treated at a time, and the process was repeated three times. After the rice leaves were inoculated at approximately 14 days, the control effect and disease index of inoculated rice leaves were evaluated.

In the same way, compound **5t** was tested for their curative activities. Thiodiazole copper and bismerthiazol were used as positive controls. After the “Fengyouxiangzhan” rice seeds were sowed at approximately one and a half months, a 2–3 cm section of rice leaf tip was cut, and the wound site was soaked in the bacterial solution for 5 s. After one day of inoculation, a solution of the compound containing 200 μg/mL of formulation was sprayed on the rice leaves, whereas 0.1% Tween-20 water was sprayed onto the negative control rice leaves. Ten leaves were also treated at a time, and the procedure was repeated three times. After the rice leaves were inoculated at approximately 14 days, the control effect and disease index of the inoculated rice leaves were measured. The degree of disease was graded as follows: level 0, no onset; level 1, lesions accounted for 1–15% of leaf area; level 2, lesions accounted for 16–30% of leaf area; level 3, lesions accounted for 31–50% of leaf area; level 4, the lesions accounted for 51–75% of leaf area; and level 5, the lesions accounted for 76–100% of leaf area.
Disease index (%) = (Σ(number of diseased plants × corresponding grade value)/(total number of diseased plants × maximum disease grade value)) × 100(2)
Control effect (%) = ((disease index of the negative control − disease index of the treatment group)/(disease index of the negative control)) × 100(3)

Statistical analysis was performed by analysis of variance (ANOVA) in SPSS 17.0 software with equal variances assumed (*p* > 0.05) and equal variances not assumed (*p* < 0.05). The different lowercase letters shown in Table 3, Table 4, Table 5 and Table 6 indicate control efficiency with significant difference among different treatment groups at *p* < 0.05.

#### 4.4.2. In Vivo Antibacterial Activities Against Rice Bacterial Leaf Streak

Potted plants were used to determine the in vivo protection and curative activity of compounds under greenhouse conditions by using the previously reported methods [32]. The target compound **5t** was assayed for protective activity against rice bacterial leaf streak. Thiodiazole copper and bismerthiazol were used as positive controls. After the “Fengyouxiangzhan” rice seeds were sowed at approximately one and a half months, the test compound in DMSO was diluted with 0.1% Tween-20 water to a concentration of 200 μg/mL. Liquid water was evenly sprayed on the rice leaves, and the negative control rice leaves were sprayed with 0.1% Tween-20 water. Then, 1/3 to 1/2 of the tips of the leaf were pierced with acupuncture containing *Xoc* by acupuncture inoculation after spraying for one day, with four holes per leaf, 10 leaves per treatment, and three replicates. After the rice leaves were inoculated at approximately 21 days, the control effect and disease index of inoculated rice leaves were evaluated.

Similarly, the target compound **5t** was assayed for curative activities against the rice bacterial leaf streak. Bismerthiazol and thiodiazole copper were used as positive controls. After the “Fengyouxiangzhan” rice seeds were sowed at approximately one and a half months, 1/3 to 1/2 of the tips of the leaf were pierced with acupuncture containing *Xoc* by acupuncture inoculation, with four holes per leaf. After one day, the dissolved test compound in DMSO was diluted with 0.1% Tween-20 water to a concentration of 200 μg/mL. The liquid was evenly sprayed on the rice leaves, whereas 0.1% Tween-20 water was used on the negative control rice. Ten leaves were treated at a time, and the procedure was repeated three times. After the rice leaves were inoculated at approximately 21 days, the control effect and disease index of inoculated rice leaves were evaluated. The degree of disease was graded as follows: level 0: no onset; level 1: lesions account for less than 1% of leaf area; level 3: lesions account for 1–5% of leaf area; level 5: lesions account for 6–25% of leaf area; level 7: lesions account for 26–50% of leaf area; level 9: lesions account for more than 50% of leaf area. The formula for calculating the disease index and control effect of rice bacterial leaf streak is the same as that of rice bacterial leaf blight.

### 4.5. Real-Time Quantitative PCR Assays

The titer of *Xoo* and *Xoc* in the rice leaves treated by compound **5t** was quantified by using Synergy Brands (SYBR) Green real-time quantitative PCR assays. The *Xoo* and *Xoc* cultures were 10-fold serially diluted from 2.0 × 10^10^ to 2.0 × 10^9^, 2.0 × 10^8^, 2.0 × 10^7^, 2.0 × 10^6^, 2.0 × 10^5^ and from 1.0 × 10^10^ to 1.0 × 10^9^, 1.0 × 10^8^, 1.0 × 10^7^, 1.0 × 10^6^, 1.0 × 10^5^, then following the manufacturer’s protocol of the TIANamp Bacteria DNA Kit (TIANGEN BIOTECH (BEIJING) CO., LTD) to extract the genomic DNA of these samples and used to produce standard curves. Genomic DNA of bacterial and leaf tissue was extracted according to the CTAB method [33]. The primers (*Xoo*230 F: 5′-CCTCTATGAGTCGGGAGCTG-3′/*Xoo*230 R: 5′-ACACCGTGATGCAATGAAGA-3′and *Xoc*112 F: 5′-CAAGACAGACATTGCTGGCA-3′/X*oc*112 R: 5′-GGTCTGGAATTTGTACTCCG-3′) and method were adopted to carry out real-time quantitative PCR assays [34]. Each PCR reaction contained 10 μL TB Green Premix Ex Taq II (Tli RNaseH Plus) (2×), 0.8 μL forward primer (10 μM), 0.8 μL reverse primer (10 μM), 2 μL template DNA and 6.4 μL ddH_2_O. The PCR program as following: 45 s at 95 °C; 40 cycles of 5 s at 95 °C, 30 s at 61 °C and melting curve at 65 °C to 95 °C with increases of 0.5 °C.

## 5. Conclusions

In this work, 31 novel 1,3,4-oxadiazole derivatives containing a cinnamic acid moiety were designed and synthesized. All of the compounds exhibited good antibacterial activities. Among them, compounds **5r** and **5t** showed better antibacterial activities against *Xoo* and *Xoc* in vitro than commercial agents thiodiazole copper and bismerthiazol. In addition, compound **5t** had good in vivo protective and curative activity against rice bacterial leaf blight and leaf streak. The 1,3,4-oxadiazole derivatives containing a cinnamic acid moiety can be used as potential new bactericides for rice bacterial diseases.

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
