# Peer review of "Novel 1,3,4-Oxadiazole Derivatives Containing a Cinnamic Acid Moiety as Potential Bactericide for Rice Bacterial Diseases"

_ijms, 2019, doi:10.3390/ijms20051020_

Round 1
Reviewer 1 Report
The authors have performed the qPCR experiment as per the earlier recommendation. The bacterial titer difference between Bismethaizol, Thiodaizole copper and 5t compound is negligible considering the standard error (I presume it is the standard error that the authors have listed in the Ct column). I do not think there is any statistical significance in the bacterial titer and the authors have clearly avoided doing a statistical analysis. The authors say average threshold cycle of three replicates: is it biological or technical replicates? How many plants did the authors analyze. The methods section of qPCR is poorly written and it doesn't give any idea about the experimental design.
The authors have not improved the quality of this manuscript despite two revisions and hence I do not recommend this manuscript for publication in IJMS.
Reviewer 2 Report
Authors have made the best efforts to respond to all concerns raised by the review to the extent possible. They have made all the changes suggested by the reviewer and performed a careful editing work.
This manuscript is a resubmission of an earlier submission. The following is a list of the peer review reports and author responses from that submission.
Round 1
Reviewer 1 Report
In this work, 31 novel 1,3,4‐oxadiazole derivatives containing a cinnamic acid moiety were well disgned and synthesized. All of the compounds exhibited good antibacterial activities. Among them, compounds 5r and 5t showed better antibacterial activities against Xoo and Xoc in vitro and in vivo than commercial agents thiodiazole copper and bismerthiazol. The only thing that needs to be improved is that you can not claim something without statistical analysis. Type of Statistical analysis isn't mentioned at all, either the Statistical program.
Author Response
In this work, 31 novel 1,3,4-oxadiazole derivatives containing a cinnamic acid moiety were well designed and synthesized. All of the compounds exhibited good antibacterial activities. Among them, compounds 5r and 5t showed better antibacterial activities against Xoo and Xoc in vitro and in vivo than commercial agents thiodiazole copper and bismerthiazol.
Answer: Thanks for your good comment on our manuscript, and your recommendation is appreciated. The manuscript has been revised according to your suggestion. We believe that the revised manuscript is more suitable for publication on International Journal of Molecular Sciences than before.
1. The only thing that needs to be improved is that you can not claim something without statistical analysis. Type of Statistical analysis isn't mentioned at all, either the Statistical program.
Answer:Thank you for your suggestions, According to your comment, Statistical analysis has been performed and the statistical results were added to the bactericidal activity test. So, in the part of “4.2. In vitro antibacterial activity”, “The results of antibacterial activities of the target compounds against Xoo and Xoc were expressed by EC50 and were evaluated by SPSS17.0 software. The experiment was repeated three times for each compound.” was added. Meanwhile, in the part of “4.3. In vivo antibacterial activity”, “Statistical analysis was performed by analysis of variance (ANOVA) in SPSS 17.0 software with equal variances assumed (P >0.05) and equal variances not assumed (P <0.05). The different lowercase letters shown in Tables 3-6 indicate control efficiency with significant difference among different treatment groups at P <0.05.” was added in the revised manuscript.
Reviewer 2 Report
The manuscript titled "Novel 1,3,4-Oxadiazole Derivatives Containing a Cinnamic Acid Moiety as Potential Bactericide for Rice Bacterial Diseases" describes the development and effectiveness of novel bactericides. The authors show the effectiveness of the bactericide and also provide a 3-D model for structure model relationship. I have the below concerns which the authors need to address.
Major concerns:
The authors have done turbidimeter assay with two concentrations of target compounds. I recommend the authors to do a agar disc diffusion assay which is a standard assay for any compound with antimicrobial property.
The authors do not describe how the disease index was calculated in the plants
The authors are recommended to quantify the bacterial titre in the invivo plants by qPCR or by bacterial growth assay. Unless they quantify the bacterial titre the invivo effectiveness of the target compound cannot be claimed just by pictorial representation
The 3-D-QSAR modelling described by the authors is very difficult to follow. The authors are recommended to make a schematic diagram to explain it better.
The figure legends do not meet the standards of IJMS. The authors are recommended to rewrite all figure legends.
The authors should distinguish figure 2 and 3 protective and curative data as Fig 2a, 2b, 3a and 3b.
The manuscripts needs extensive English editing, I recommend the authors take the help of a native English speaker to correct the English
Minor comments:
Line 36: change care to are
Line 41: these two bacterial diseases ----which bacterial diseases
Line 229: A solvent -----which solvent
Line 255: solvent NB media----what does the authors mean by solvent NB medi
Author Response
The manuscript titled "Novel 1,3,4-Oxadiazole Derivatives Containing a Cinnamic Acid Moiety as Potential Bactericide for Rice Bacterial Diseases" describes the development and effectiveness of novel bactericides. The authors show the effectiveness of the bactericide and also provide a 3-D model for structure model relationship. I have the below concerns which the authors need to address.
Answer: Thanks for your good comment on our manuscript, and your recommendation is appreciated. The manuscript has been revised according to your suggestion. We believe that the revised manuscript is more suitable for publication on International Journal of Molecular Sciences than before.
1. The authors have done turb idimeter assay with two concentrations of target compounds. I recommend the authors to do a agar disc diffusion assay which is a standard assay for any compound with antimicrobial property.
Answer:Thank you for your suggestions. Both turbidimeter assay and agar disc diffusion assay can be used for evaluation the antimicrobial property of compound. The agar disc diffusion assay, also named disk diffusion test, or agar diffusion test, or Kirby–Bauer test (disc-diffusion antibiotic susceptibility test, disc-diffusion antibiotic sensitivity test, KB test), is a test of the compound sensitivity to bacteria. It uses antibiotic discs to test the extent to which bacteria are affected by compound. In this test, wafers containing antibiotics are placed on an agar plate where bacteria have been placed, and the plate is left to incubate. Then, the antibacterial activity of compound was evaluated with zone of inhibition (Antibacterial evaluation of some schiff bases derived from 2-acetylpyridine and their metal complexes. Molecules, 2012, 17, 5952−5971; Principles of assessing bacterial susceptibility to antibiotics using the agar diffusion method. J. Antimicrob. Chemoth. 2008, 61, 1295−1301.). The process of the method included preparation of plate, inoculation with pathogen, incubation and determination of zone of inhibition. There are many operational factors that lead to experimental errors. The turbidimeter assay is the process of measuring the loss of intensity of transmitted light due to the scattering effect of particles suspended in it. The advantage of this method is that the Nutrient broth (NB) media was added with compound and pathogens, and together incubated, the optical density at 595 nm (OD595) was measured (Estimation of bacterial growth rates from turbidimetric and viable count data .Int. J. Food Microbiol. 1994, 23,391−404; Inhibition of tobacco bacterial wilt with sulfone derivatives containing an 1,3,4-oxadiazole moiety. J. Agric. Food Chem. 2012, 60, 1036−1041.). Compared with the agar disc diffusion assay, the turbidimeter assay is fast and easy to operate, and the results are objectivity and precision which were measured by instruments, and the sensitivity of the test is high using liquid culture method, without the influence of diffusion factors. At present, the disc diffusion method is mainly used for bactericidal determination in medicine, and is less used in pesticides. The turbidimeter assay was widely used to evaluating the antimicrobial property of pesticide. (Design, synthesis, and evaluation of new sulfone derivatives containing a 1,3,4-oxadiazole moiety as active antibacterial agents. J. Agric. Food Chem. 2018, 66, 3093−3100; Photochemical degradation of bismerthiazol: structural characterisation of the photoproducts and their inhibitory activities against Xanthomonas oryzae pv. oryzae. Pest Manage. Sci. 2016, 72, 997−1003). To sum up, the turbidimeter assay was used to test antimicrobial activity of compound.
2. The authors do not describe how the disease index was calculated in the plants.
Answer: Thank you for your suggestions. We are sorry that we did not describe the method for evaluating disease index in the plants. According to your comment, the calculated methods of the disease index in the plant were added in the revised manuscript, as follows:
In the part of “4.3.1. In vivo antibacterial activities against rice bacterial leaf blight”, “The degree of disease was graded as follows: level 0, no onset; level 1, lesions accounted for 1%-15% of leaf area; level 2, lesions accounted for 16%-30% of leaf area; level 3, lesions accounted for 31%-50% of leaf area; level 4, the lesions accounted for 51%-75% of leaf area; and level 5, the lesions accounted for 76%-100% of leaf area.
Disease index (%) = [Σ (number of diseased plants × corresponding grade value) / (total number of diseased plants × maximum disease grade value)] × 100
Control effect (%) = [(disease index of the negative control - disease index of the treatment group) / (disease index of the negative control)] × 100” was added.
Meanwhile, in the part of “4.3.2. In vivo antibacterial activities against rice bacterial leaf streak”, “The degree of disease was graded as follows: level 0: no onset; level 1: lesions account for less than 1% of leaf area; level 3: lesions account for 1%-5% of leaf area; level 5: lesions account for 6%-25% of leaf area; level 7: lesions account for 26%-50% of leaf area; level 9: lesions account for more than 50% of leaf area. The formula for calculating the disease index and control effect of rice bacterial leaf streak is the same as that of rice bacterial leaf blight.” was added in the revised manuscript.
3. The authors are recommended to quantify the bacterial titre in the in vivo plants by qPCR or by bacterial growth assay. Unless they quantify the bacterial titre the in vivo effectiveness of the target compound cannot be claimed just by pictorial representation.
Answer:Thank you for your suggestions. Bacterial blight of rice caused by Xanthomonas oryzae pv. oryzae. The rice bacterial blight pathogen invades through wounds or hydathode water pores to gain access to the plant’s xylem vessels, where it multiplies to plug the vessels. Bacterial leaf streak of rice caused by Xanthomonas oryzae pv. oryzicola. The pathogen invades through wounds or stomates, and moves and lives between the mesophyll parenchyma cells of the rice leaves. (Novel insights into rice innate immunity against bacterial and fungal pathogens. Annu. Rev. Phytopathol.2014, 52, 213−241.). Yellow spots appear after the leaves are infected with two bacterial diseases. These spots gradually become larger and yellow, and gradually spread upward as the rice plants develop. The lesions are usually begins at the edge of the upper part of the leaf, where the pores in which the bacteria can invade are more frequently distributed. Then, the lesion expands along the vein and turns yellow. (Epidemiology and control of bacterial leaf blight of rice. Annu. Rev. phytopathol. 1969, 7, 51-72; An improved infiltration technique to test the pathogenicity of Xanthomonas oryzae pv. oryzae in rice seedlings. Seed Sci. Technol.1996, 24, 449−456.). These enlarged yellow lesions will later turn white or off-white. Therefore, the control effect can be initially obtained by measuring the length of the leaf lesion. (Design, synthesis, and evaluation of new sulfone derivatives containing a 1,3,4-oxadiazole moiety as active antibacterial agents. J. Agric. Food Chem. 2018, 66, 3093−3100. Inhibition of tobacco bacterial wilt with sulfone derivatives containing an 1,3,4-oxadiazole moiety. J. Agric. Food Chem. 2012, 60, 1036−1041.). Through the external observation and determination of the length of the lesion, the activity of the agent on the bacterial disease is preliminarily judged. Quantification bacterial titre in the in vivo plants by qPCR or by bacterial growth assay is fast, objective, precision, and high sensitive method. According to your comment, the method of quantification the bacterial titre in the in vivo plants by qPCR or by bacterial growth assay is used to evaluate the antimicrobial property in our subsequent studies.
4. The 3-D-QSAR modelling described by the authors is very difficult to follow. The authors are recommended to make a schematic diagram to explain it better.
Answer: Thank you for your suggestions. The manuscript has made a schematic diagram for 3D-QSAR to explain it better. So, the following sentence and Figure 8 were added “In addition, the 3D-QSAR model showed satisfactory predictive ability, and the results illuminated (as shown in Figure 8) that bulky groups, negatively charged groups, hydrophilic groups at R1 position, and small groups, positive groups, hydrophobic groups, H-bond acceptor groups at the R2 position favor activity. Meanwhile the sulfone group is the essentially active group. These results provide further valuable clues for the design of novel 1,3,4-oxadiazole derivatives containing a cinnamic acid moiety.Figure 8. The structure–activity relationship (SAR) summarized based on our work”
5. The figure legends do not meet the standards of IJMS. The authors are recommended to rewrite all figure legends.
Answer: Thank you for your suggestions. The manuscript has been reworked with all figure legends. We believe that the revised manuscript is more suitable for publication on International Journal of Molecular Sciences than before.
6. The authors should distinguish figure 2 and 3 protective and curative data as Fig 2a, 2b, 3a and 3b.
Answer: Thank you for your suggestions. Figure 2 and 3 protective and curative data were distinguished as Fig 2a, 2b, 3a and 3b, as follows:
Figure 2. Protective (a) and curative (b) activities of compounds 5t against rice bacterial leaf blight.
Figure 3. Protective (a) and curative (b) of compounds 5t against rice bacterial leaf streak.
7. The manuscripts needs extensive English editing, I recommend the authors take the help of a native English speaker to correct the English
Answer: Thank you for your suggestions. According to your comment, the manuscript has been checked by a professional English editing service from MDPI (english-edited-6750) for perfection in English writing and grammars, these revisions are colored red in the revised manuscript and also listed in part of other corrections made by authors. We believe that the revised manuscript is more suitable for publication on International Journal of Molecular Sciences than before.
8. Line 36: change care to are
Answer:Thank you for your suggestions. The “care” was changed to “are”.
9. Line 41: these two bacterial diseases ----which bacterial diseases
Answer:Thank you for your mention. The “these two bacterial diseases” was changed to “rice bacterial leaf blight and leaf streak”.
10. Line 229: A solvent -----which solvent
Answer:Thank you for your mention. we regretted this mistake on expression, “A solvent was slowly added dropwise with carbon disulfide (0.75 mol) at room temperature to the three-necked flask containing KOH (0.6 mol) in ethanol (500 mL)…” was changed to “Carbon disulfide (0.75 mol) was slowly dropwise added to the three-necked flask containing KOH (0.6 mol) of ethanol (500 mL) at room temperature…”.
11. Line 255: solvent NB media----what does the authors mean by solvent NB media
Answer:Thank you for your mention. The “…solvent NB media having the pathogens of …” was changed to “…culture solution containing the pathogens of…”.

Round 2
Reviewer 2 Report
The authors have just paid lip service rather than make any significant revisions to the manuscript. Authors agreeing to do qPCR in their future publications doesn't address the need of the current manuscript submitted to IJMS. The authors response clearly suggest that the authors did not take the recommendation seriously. If time was a factor they should have requested the editor for additional time and completed the experiment or should have rebutted why a qPCR quantification is not necessary and how their methods are better. Acknowledging the importance of bacterial quantification but not doing it is not acceptable.